# A QM/MM Study on the Initiation Reaction of Firefly Bioluminescence—Enzymatic Oxidation of Luciferin

**DOI:** 10.3390/molecules26144222

**Published:** 2021-07-12

**Authors:** Mohan Yu, Yajun Liu

**Affiliations:** 1Center for Advanced Materials Research, Advanced Institute of Natural Sciences, Beijing Normal University at Zhuhai, Zhuhai 519087, China; tmsdtkl@foxmail.com; 2School of Chemistry and Chemical Engineering, Liaoning Normal University, Dalian 116029, China; 3Key Laboratory of Theoretical and Computational Photochemistry, Ministry of Education, College of Chemistry, Beijing Normal University, Beijing 100875, China

**Keywords:** firefly bioluminescence, luciferin oxidation, mechanism, single electron transfer, QM/MM

## Abstract

Among all bioluminescent organisms, the firefly is the most famous, with a high luminescent efficiency of 41%, which is widely used in the fields of biotechnology, biomedicine and so on. The entire bioluminescence (BL) process involves a series of complicated in-vivo chemical reactions. The BL is initiated by the enzymatic oxidation of luciferin (LH_2_). However, the mechanism of the efficient spin-forbidden oxygenation is far from being totally understood. Via MD simulation and QM/MM calculations, this article describes the complete process of oxygenation in real protein. The oxygenation of luciferin is initiated by a single electron transfer from the trivalent anionic LH_2_ (**L**^3−^) to O_2_ to form ^1^[**L**^•2−^…O_2_^•−^]; the entire reaction is carried out along the ground-state potential energy surface to produce the dioxetanone (FDO^−^) via three transition states and two intermediates. The low energy barriers of the oxygenation reaction and biradical annihilation involved in the reaction explain this spin-forbidden reaction with high efficiency. This study is helpful for understanding the BL initiation of fireflies and the other oxygen-dependent bioluminescent organisms.

## 1. Introduction

The firefly is the most efficient bioluminescent system for converting chemical energy into light with the extremely high luminescence efficiency of 41% [1]. Its bioluminescence (BL) has been applied widely in biotechnology and biomedical fields [2,3]. The entire firefly BL process can be roughly divided into four stages (Figure 1) [4,5,6,7]: oxidation of luciferin (LH_2_) to a dioxetanone (FDO), decomposition of the dioxetanone to produce excited-state oxyluciferin (bioluminophore), fluorescence emission [8,9,10] and LH_2_ regeneration. The mechanism of the last three stages has been explained theoretically in detail [11,12,13,14,15,16], but there is no comprehensive and reliable theoretical study on the mechanism of the first stage (blue dotted box in Figure 1). Oxygenation is the initial reaction for not only firefly BL but also all oxygen-dependent BL systems. A thorough and reliable investigation of this process is of great significance for understanding the mechanism of all oxygen-dependent BL [17,18,19,20].

In 2015, Branchini et al. detected the presence of superoxide anion (O_2_^•−^) in the chemical model reaction of firefly BL and suggested that firefly BL is induced by single electron transfer (SET) from LH^−^ to oxygen [21]. There are some corresponding theoretical studies on the oxygenation of luciferin. In 2018, an umbrella sampling molecular dynamics simulation and QM/MM study pointed out the approach of the oxygen moving inside the protein and defined the formation of FDO^−^, but did not provide details along the reaction path [22]. Our group has investigated the oxygenation process in DMSO, and described a complete process with potential energy curves (PECs) of both ground state (S_0_) and triplet state (T_1_) to confirm the SET mechanism [23]. However, this calculation was not performed in the real protein, and the conclusion could not reflect the essence of the enzymatic reaction. Although there is experimental evidence and corresponding theoretical calculations, the below questions have not yet been thoroughly answered. What is the entire reaction process from L^2−^-AMP + O_2_^•−^ to FDO^−^ in protein? What is the difference between the oxygenation pathway in a solvent and in protein? How does the spin-forbidden reaction of firefly BL occur so efficiently? To answer these three questions is the purpose of this article.

## 2. Computational Details

For the LH_2_, the H atom on the hydroxyl in benzothiazole moiety of luciferin is easy to lose in the luciferase environment (Scheme 1). Besides, it has been proven that the dioxetanone decomposition is caused by FDO in its anionic form (FDO^−^). In addition, the H atom on C_4_ site of LH-AMP is removed by the adjacent residue in luciferase; this process has already been verified by theoretical calculation [22]. Therefore, the complex L^2^^−^-AMP (for convenience, this trivalent complex is named **L**^3^^−^) is the actual reactant in this study. **L**^2^^−^ and O_2_^•^^−^ are produced from SET by **L**^3^^−^ to O_2_, which induces the subsequent superoxide anion addition reaction. The North American firefly *Photinus pyralis* luciferase (PDB ID 4G37) [24] was chosen for its structure, which is suitable for providing a starting point for simulating the oxygenation reactions. The 2.5 ns molecular dynamic (MD) simulation was performed to consider protein fluctuation. The initial structures for the QM/MM calculations were started at the snapshot of 1800 ps from the MD trajectory (Appendix A). The chosen QM region contains **L**^3−^ and O_2_ with a total of 59 atoms, as shown in Figure 2. All QM/MM calculations were performed by a two-layered ONIOM method encoded in the Gaussian16 program [25]. The UM06-2X [26]/6-311G (d, p) [27,28] method with broken-symmetry technology was adopted for the QM region, and the remainder of the system (MM region) used the Amber force field (parm96). The QM/MM calculations and the MD simulation were based on the Gaussian 16 package [25] and AMBER 16 [29], respectively. Computational details are given in the Appendix A.

## 3. Results and Discussion

The firefly oxygen addition of **L**^3−^ is initiated by a SET process from **L**^3−^ to ^3^O_2_ to produce two free doublet radicals, **L**^•2−^ and O_2_^•−^; this reactant complex (RC) has been experimentally [21] confirmed. Since the RC is a biradical ionic pair formed by **L**^•2−^ and O_2_^•−^ radicals, ^3^[**L**^•2−^…O_2_^•−^] and ^1^[**L**^•2−^…O_2_^•−^] are both possible initial states. For the spin density on the atom for ^3^[**L**^•2−^…O_2_^•−^] and ^1^[**L**^•2−^…O_2_^•−^], see Appendix A. Regarding ^1^[**L**^•2−^…O_2_^•−^] as the initial state, from RC to the final product FDO^−^ and AMP (for convenience, we defined FDO^−^ + AMP as P), three transitions states (TSs) and two intermediates (Ints) were located. The relative energy profiles of SET oxygenation on **L**^3−^ are shown in Figure 3. The key geometric parameters and Mulliken charges population for all stationary points on **L**^•2−^ and O_2_^•−^ moieties are summarized in Table 1. As shown in Figure 3, the RC ^1^[**L**^•2−^…O_2_^•−^] (^1^RC) is formed by electrostatic force and van der Waals interaction. The O_a_-O_b_ bond distance is 1.311 Å in ^1^RC, which is longer than it is in ^3^O_2_ (1.205 Å). This implies that O_2_^•−^, rather than ^3^O_2_, attacks the **L**^•2−^. The charge distribution of O_2_^•−^ is −1.00 |e| when it is just generated via the SET process. However, partial negative charge transferred from O_2_^•−^ to **L**^•2−^ in the formation of ^1^RC, and the Mulliken charge on O_2_^•−^ is −0.88 |e| and −2.12 |e| on **L**^•2−^. The expectation value of the *S*^2^ operator (<*S*^2^>) is 1.00, which indicates that the ^1^RC has obvious biradical characteristics. With the process of nucleophilic addition, the C_4_-O_a_ bond becomes shorter and the C_6_-C_4_-O_a_-O_b_ dihedral angle gradually twists to the closure of four-membered cyclic peroxide. ^1^RC forms **Int1** through **TS1** via a biradical annihilation. This process is accompanied by a small amount of back negative charge transfer (CT) from **L**^•2−^ to O_2_^•−^ (see Table 1). The bond length of C_4_-O_a_ is 1.401 Å in **Int1**, which does not change much until **TS2**. **Int1** forms **Int2** with a four-membered cyclic structure through **TS2**.

The main structural changes from **Int1** to **Int2** are the shortening of the C_6_-O_b_ bond and the torsion of the C_6_-C_4_-O_a_-O_b_ dihedral angle, which are accompanied by an obvious negative CT from O_2_^•−^ to **L**^•2−^ (see Table 1). The bond length of C_6_-O_8_ does not change much until **Int2**. After **Int2**, the C_6_-O_8_ bond began to break and leads to **P** with the departure of the AMP group. This process is accompanied by −0.17 CT from O_2_^•−^ to **L**^•2−^ and adjustment of the C_6_-C_4_-O_a_-O_b_ dihedral angle. (See Table 1).

For the case of ^3^[**L**^•2−^…O_2_^•−^] as the initial state, the energies of all stationary points at the T_1_ state were evaluated at the corresponding S_0_ geometries, except ^3^RC was optimized (Appendix A). The optimized ^3^[**L**^•2−^…O_2_^•−^] is 0.7 kcal mol^−1^ higher than ^1^[**L**^•2−^…O_2_^•−^]. The ^3^[**L**^•2−^…O_2_^•−^] at the ^1^[**L**^•2−^…O_2_^•−^] geometry is 1.6 kcal mol^−1^ higher than ^1^[**L**^•2−^…O_2_^•−^]. Except for ^3^[**L**^•2−^…O_2_^•−^], the energy of each stationary point on the T_1_ PEC is much higher than the corresponding one on the S_0_ PEC. Obviously, the LH_2_ oxygenation reaction occurs on the S_0_ potential energy surface (PES). It is worth mentioning that this is quite different from our previous calculation in DMSO [23]. In a solvent, the reaction first occurs along the T_1_ PES; after an intersystem crossing (ISC), the reaction takes place on the S_0_ PES. Meanwhile, the biradical annihilation occurs along with the ISC process and finally a four-membered cyclic structure is formed (Appendix A). However, in luciferase, the reaction always occurs on the S_0_ PES and does not need to pass an ISC, which definitely increases the reaction efficiency. Along the reaction process, the highest energy barrier is 7.7 kcal mol^−1^, which is lower than the highest energy barrier of 11.9 kcal mol^−1^ in a solvent (Appendix A).

To further probe the effect of enzymatic catalysis in the process of LH_2_ oxidation, we carefully investigated the important interactions between the QM region and the residues in active-site residues (see Figure 4). According to the above computational results in the protein, the rate-determined step in the complete process of the LH_2_ oxygenation is from ^1^RC to **Int1**. Therefore, our analysis focuses on the structures of ^1^RC, **TS1** and **Int1**. F-shaped π-π stacking forming between Phe-247 and **L**^•2−^ is important for stabilizing the aromatic part of the benzothiazole moiety of **L**^•2−^; this interaction exists all throughout the oxygenation. Besides, His-245 and Lys-433 had a strong H-bond interaction with the O atom on the AMP moiety of **L**^3−^. Lys-433 and O_7_ of **L**^3−^ also formed an H-bond. These H-bond interactions between positively charged residues and substrates lead to the negative charge of O_2_^•−^ and **L**^•2−^ moiety decrease. The H-bond interaction with 2.453 Å between Gly-246 and O_2_^•−^ affects the relative position of O_2_^•−^ and **L**^•2−^. With the C_6_-C_4_-O_a_-O_b_ dihedral angle gradual torsion, the H-bond between Gly-246 and O_2_^•−^ becomes slightly strong, and then changes to 2.647 Å as **Int1** is formed. Meanwhile, His-245 with O_b_ forms a weak H-bond. In short, these results demonstrated that the hydrogen-bond interactions between Gly-246 and O_2_^•−^ are essential for the process of oxygen addition. The hydrogen-bonding interactions between Lys-433 and His-245 with AMP moiety of **L**^3−^ majorly stabilize the substrate and the negative charge on the O atom of AMP.

## 4. Conclusions

Firefly BL is initiated by the reaction of LH_2_ + ^3^O_2_. This is a spin-forbidden reaction with usually a low efficiency, which contradicts the fact that the firefly is the most efficient bioluminescent system for converting chemical energy into light. In this letter, we addressed this issue via MD and QM/MM studies. In luciferase, the entire reaction starts with a SET process and occurs all along the S_0_ PES. This is quite different for the reaction in a solvent, where the reaction first occurs along the T_1_ PES and then along the S_0_ PES after an intersystem crossing (ISC). Moreover, the rate-determined step obviously has a lower barrier in luciferase than in a solvent. The effect of enzymatic catalysis was analyzed. The present theoretical study provides strong theoretical evidence for the SET mechanism of LH_2_ oxygenation of firefly BL, and is helpful for understanding the BL initiation of the other oxygen-dependent bioluminescent organisms.

## Data Availability

Data are available from the authors.

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
