# Peer review of "A QM/MM Study on the Initiation Reaction of Firefly Bioluminescence—Enzymatic Oxidation of Luciferin"

_molecules, 2021, doi:10.3390/molecules26144222_

Round 1
Reviewer 1 Report
The authors present an interesting work on the formation of the oxyluciferin precursor. Using QM/MM methodology and potential energy surface analysis, they obtain the reaction mechanisms for the two possible spin multiplicities: single and triplet states. Unlike what happens in the solution, where there is an initial triplet-singlet conversion, the proposed mechanism in the enzyme takes place entirely in the singlet state.
concerns:
- In Scheme 1, wouldn't there be extra hydrogen at the carboxyl group of the substrate?
- It would be interesting if you could quote figure S3 at the beginning of section 2, to facilitate the numbering of the atoms.
- It would be very useful to have the spin densities for some of the atoms, especially in the case of the triplet.
- In section 1.3 of the supplementary material instead of HIP it should be HIE.
- Any indication why they specifically chose the snapshot corresponding to 1.8 ns?
- I have not checked all of them, but there is a missing negative sign in the Y coordinate of the O atom (10.489219) for RC in T1.
- PDB has some residues missing in the middle... How has this been managed by authors when using H++?
The main criticism of the work done is in the triplet/singleton discussion. Except for the first stage of the mechanism, the triplet state is naturally disfavored, since it would be located on the C6-O7 carbonyl, which is expected to have higher energy than the single. Although it will not change the main conclusions (the singlet state is always the most stable state in the enzyme), the authors' point calculations on singlet geometries will not help, since in singlet geometry, the C=O double bond will be properly defined.
Reviewer 2 Report
In this work, Yu et al. employed a computational approach to provide insight into an initial reaction of the firefly bioluminescence process – an oxidation of luciferin to a dioxetanone. For several years, the intensive studies have been carried out on the mechanism of bioluminescence reaction of various living organisms, especially the bioluminescence of firefly. However, these studies focused mainly on the formation of the electronically excited products and then the emission of radiation (fluorescence). The first stage of the bioluminescence process is also important and should be carefully studied. An investigation of the luminescence processes is one of the topics of photochemical studies, so the work fits the current trends in photochemistry well. The work presents interesting research results. It is also well written. The calculations have been done carefully and no obvious errors or omissions could be detected by referee. Therefore, I can recommend publication after minor revision.
- In my opinion, the Authors introduce the subject of the bioluminescence process of firefly very precisely. However, the literature cited is very selected. In recent years, studies on the bioluminescence process have been conducted intensively, especially by Lindh’s research group or da Silva’s research group. I recommend the following articles: 10.1002/cphc.201100504, 10.1016/j.cplett.2014.05.061, 10.1002/cphc.201200195, 10.1021/ja908051h, 10.1021/ct200045q, 10.1002/cphc.201100389.
- The oxygenation pathway differs in solvent (DMSO) than in protein. How does the Authors explain that? Is the solvent (DMSO) properly selected for the studied the mechanism of bioluminescence process?
- The Authors use the Gaussian16 package for QM/MM calculations. The version of Gaussian16 revision should be given in the references.
